# Risk Assessment of (Herbal) Teas Containing Pyrrolizidine Alkaloids (PAs) Based on Margin of Exposure Approach and Relative Potency (REP) Factors

**DOI:** 10.3390/foods11192946

**Published:** 2022-09-21

**Authors:** Lu Chen, Qian Zhang, Ziwei Yi, Yu Chen, Weihan Xiao, Dan Su, Wenbiao Shi

**Affiliations:** 1Department of Nutrition and Health, Chinese Agriculture University, Beijing 100091, China; 2Institute of Quality Standards and Testing Technology for Agro-Products, Chinese Academy of Agricultural Sciences, Key Laboratory of Agri-food Quality and Safety, Ministry of Agriculture and Rural Affairs, Beijing 100081, China; 3Department of Chemistry and Chemical Biology, Cornell University, Ithaca, NY 14850, USA

**Keywords:** herbal tea, pyrrolizidine alkaloids (PAs), margin of exposure (MOE), risk assessment, relative potency (REP)

## Abstract

Pyrrolizidine alkaloids (PAs) present distinct toxicity potencies depending on their metabolites and in vivo toxicokinetics. To represent the potency differences of various PAs, the interim relative potency (REP) factors have been derived. However, little is known about the risk assessment for (herbal) teas when taking REP factors into account. In this study, a set of 68 individual 1,2-unsaturated PA in 21 types of (herbal) teas was analyzed using LC-MS/MS. The REP factors for these PAs were applied on the PA levels. The margin of exposure (MOE) approach was employed to assess the risks of the exposure to PAs due to consumption of (herbal) teas. The results show that the total PA levels ranged from 13.4 to 286,682.2 μg/kg d.m., which were decreased by REP correction in most of the teas. The MOE values for tephroseris, borage and lemon balm (melissa) tea based on REP-corrected PA levels were below 10,000, assuming daily consumption of one cup of tea during a lifetime, indicating that consuming these teas may raise a concern. Our study also indicates a priority for risk management for tephroseris tea, as having nephrosis tea for more than 11.2 weeks during a 75-year lifetime would result in an MOE of 10,000.

## 1. Introduction

Pyrrolizidine alkaloids (PAs) are toxic substances that exist naturally in plants [1]. To date, over 660 types of PAs and PA *N-*oxide have been identified in the estimated six thousand plants [2]. 1,2-unsaturated PAs are particularly of concern as they are hepatotoxicants and genotoxic carcinogens [3]. 1,2-unsaturated PAs can be subdivided by the type of esterification, including monoesters, open chained diesters, and cyclic diesters. In addition, cyclic diester PAs with an azacyclooctanone, instead of a 1,2-dehydropyrrolizidine ring system, represent a special class. The main human exposure route to PAs is consuming plant-derived foods, such as (herbal) teas. PAs can induce hepatotoxicity both in humans and animals [1]. Human poisoning and even deaths from the consumption of PAs have been reported in several countries [4]. Severe outbreaks of the PA contamination once occurred in Afghanistan [5] and central India [6]. In addition, a few cases were reported in Hong Kong [7], Switzerland [8,9], Austria [10] and Tajikistan [11]. As a result, the use of the PA-containing plants as food products or supplements has been restricted in several countries [12]. However, there is still a lack of global consensus in regulatory measures regarding PAs in plant-derived products so far. Possibly, conducting a risk assessment for foodstuff containing PAs may contribute to the development of such a consensus.

(Herbal) teas are a type of PA-containing plant-derived product. The concentration of PAs can vary enormously among (herbal) teas. For instance, the Federal Institute for Risk Assessment (BfR) analyzed 274 types of (herbal) teas and found that the levels of PAs ranged from below the level of detection (LOD) to 5647.2 μg/kg dry material (d.m.) [2]. Later, Mulder et al. (2015) reported that the concentration of PAs could reach up to 4804.5 µg/kg d.m. based on the analyzed 22 types of (herbal) teas [13].

Due to the omnipresence of PAs in (herbal) teas and their detrimental effects, the safety evaluation of PAs associated with the consumption of teas is crucial. Multiple studies have performed risk assessments for PAs in (herbal) teas in the recent years [3,14,15]. Given that 1,2-unsaturated PAs are genotoxic and carcinogens, the risk assessment was conducted based on the margin of exposure (MOE) approach [16]. The MOE is defined as the ratio between the benchmark dose level with a lower confidence limit associated with a 10% extra risk on a cancer incidence above background levels (BMDL_10_) and the estimated daily intake (EDI). To date, the BMDL_10_ values for only two PAs, lasiocarpine and riddelliine, have been derived, whereas the values for the other PAs remained unavailable due to the lack of appropriate animal carcinogenicity studies [17,18]. The European Food Safety Authority (EFSA) used to adopt a BMDL_10_ of 0.07 mg/kg bw/day of lasiocarpine as the point of departure (PoD) for the MOE calculation [19]. A cut-off value of 10,000 for the MOE is usually applied, which incorporates factors including the inter-species and intra-species differences in toxicokinetic and toxicodynamics, the inter-individual human variability in cell cycle control and DNA repair as well as the potential discrepancy between the BMDL_10_ serving as a reference point and a NOAEL (No Observed Adverse Effect Level) [16]. The risk assessment suggested that the long-term consumption of several (herbal) teas may pose a potential health risk in humans, especially when considering a lifetime exposure [3,14,15]. It is worthwhile to mention that the MOE values were calculated based on the mean levels of total PAs in these studies, assuming that the metabolism and toxic potencies of PAs were the same with lasiocarpine. This, however, may result in an overestimation of potential risks from the exposure, since the toxic potencies of individual PAs are distinct and most of them could be lower than lasiocarpine. In addition, the obtained BMDL_10_ value for lasiocarpine (70 μg/kg bw/day) was affected by a high degree of uncertainty [20]. Instead, the EFSA has proposed the BMDL_10_ of riddelliine, which was 237 μg/kg bw/day, for the combined risk assessment of PAs by dose addition. The relative potency factor (REP) correction serves as an approach for the risk assessment for a mixture of chemicals that exhibit a common mode of action. To derive a REP, the potency of each component in a mixture is compared to that of a reference chemical generating a measure of potency for each component with respect to the toxicity of the index chemical [12]. It is more rigorous to perform risk assessments for PA-containing botanicals by determining the REP factor for each PA contained and then adjusting the concentration of each PA in the mixtures for the assessments.

To date, little is known about risk assessments for PAs in botanical samples when taking into account the REP factor of each PA. Additionally, the BMDL_10_ of riddelliine was proposed as the new PoD for the MOE calculation. Considering these facts, the actual exposure and related risk assessment of PAs due to the consumption of (herbal) teas need to be reevaluated. Therefore, the aim of the present study is to determine the PA levels in 21 types of (herbal) teas with and without the correction of REP factors, and to perform risk assessments based on the BMDL_10_ of riddelliine derived MOE approach. To achieve this, in total 68 individual 1,2-unsaturated PAs were analyzed, including cyclic diesters and heliotridine-type (7S) open diesters (e.g., monocrotaline, retrorsine, riddelliine, senecionine, seneciphylliine, senkirkine, heliosupine and lasiocarpine), heliotridine-type (7S) monoesters (e.g., echinatine and heliotrine), retronecine-type (7R) open diesters (e.g., echimidine and symphytine), and retronecine-type (7R) monoesters (e.g., indicine, intermedine and lycopsamine) monoesters, open chained diesters and cyclic diesters. The chemical structure for these PAs and their *N-*oxide form, as well as their corresponding REP factors, has been previously reported in detail [12]. To clarify the rationale for this study, a graphical workflow is shown as Figure 1.

## 2. Materials and Methods

### 2.1. PA Extraction

Twenty-one types of commercial (herbal) teas were sampled from China and EU countries, including 7 types of teas derived from PA-containing plants and 14 types from non-PA producing plants. Detailed information of the (herbal) teas is listed in Table 1. In total, 147 tea samples, which were present as comminuted leaves, were analyzed. Two grams of (herbal) tea were weighed out and transferred to beaker glasses or brown bottles. Immediately, 150 mL of boiling water was added onto the samples. The infusion was steeped for 10 min, with a gentle stirring for 10 s at 0, 3 and 6 min, respectively. The infusion was allowed to cool down at room temperature, after which it was filtered by passing through a 0.45 µm filter. Fifty milliliters of the infusion were subject to the following cleanup procedure.

### 2.2. Sample Cleanup

The sample cleanup was carried out using reversed phase C18 solid phase extraction (SPE) cartridges (Discovery DSCC18 500 mg/5 mL, Supelco, Bellefonte, PA, USA), as described previously [21]. The SPE cartridges were conditioned using 5 mL of methanol and water, respectively. The cartridges were loaded with 50 mL of the filtered infusion, washed with 6 mL of water and then dried with a vacuum manifold for 10 min. The elution of the PAs for all the tea samples was done by adding 5 mL of methanol. The elution step was repeated. The eluates were combined, dried under a gentle nitrogen stream in a warmed water bath (50 °C, TurboVap, Biotage, Uppsala, Sweden) and reconstituted in 1 mL of methanol/water (5/95, *v*/*v*). The extracts were transferred into filter columns (Nylon, 0.2 μm, VWR, Darmstadt, Germany) and centrifuged at 13,000× *g* for 2 min at room temperature.

### 2.3. LC-MS/MS Analysis

The PAs in the samples were measured using a LC-MS/MS system consisting of an UHPLC (Ultimate 3000, Thermo Scientific, San Jose, CA, USA) and a Triple Stage Quadrupole mass spectrometer (TSQ Vantage, Thermo Scientific, San Jose, CA, USA), as described previously with minor modifications [21]. Briefly, chromatographic separation was achieved on 150 × 2.1 mm, 1.9 μm particle sizes, C18 Hypersil Gold column fitted with a guard column (Thermo Scientific, Dreieich, Germany). Eluent A was 100% water with 0.1% formic acid and 5 mM of ammonium formate. Eluent B was 95% methanol and 5% water with 0.1% formic acid and 5 mM of ammonium formate. A stepwise gradient elution was conducted as follows: 0–0.5 min for 95% A/5% B, 7.0 min for 50% A/50% B, 7.5 min for 20% A/80% B, 7.6–9.0 min for 100% B and 9.1–15 min for 95% A/ 5% B. A flow rate of 300 μL/minute was applied and 10 μL of each sample was injected. The column temperature was maintained at 40 °C. Details of the mass parameters are listed in Appendix A.

### 2.4. Quality Assurance and Quality Control (QA/QC)

Fity-four PA standard compounds were obtained from the following sources: echimidine, indicine, indicine *N-*oxide, intermedine, intermedine *N-*oxide, lycopsamine, lycopsamine *N-*oxide, monocrotaline, monocrotaline *N-*oxide and otosenine from Phytolab (Vestenbergsgreuth, Germany); heliotrine and trichodesmine from Latoxan (Valence, France); usaramine from BOC Sciences (Shirley, Suffolk, NY, USA); florosenine from PRISNA (Leiden, the Netherlands); and Usaramine *N-*oxide and trichodesmine *N-*oxide were in-lab synthesized according to [22] and the rest from Phytoplan (Heidelberg, Germany). An analytical grade of formic acid and ammonium carbonate (Energy Chemical) and a LC-MS grade of acetonitrile and methanol (Sinopharm) were purchased from Shanghai, China.

Quality assurance and quality control (QA/QC) procedures were performed as follows: a procedural blank, a spike blank, a mixed PA sample (1 μg/mL of external standards in methanol, used to spike the (herbal) teas in 25 ng/mL and 100 ng/mL) and a duplicate were run for each batch of 20 samples to check for cross-contamination and instrumental reliability as well as to indicate recoveries. No PAs in the blanks were detected. The standard deviations for standard solution (7-point calibration curves over the range of zero to 250 ng/mL) were controlled within 10%. Recoveries at the level of 100 ng/mL varied from 79 to 110%. The LOD for PAs in the infusion of (herbal) teas was determined as the concentrations of analyses in a sample that showed a peak divided by the signal-to-noise ratio (S/N) of 3. The LOD was estimated at the range of 10 to 20 ng/L and the limit of quantification (LOQ) was obtained at 50 ng/L. For those PAs with no available standard compounds, the corresponding structurally related PA standards were employed for the semi-quantification of those PA levels.

### 2.5. Estimated Daily Intake (EDI) and MOE Calculation

The EDIs of PAs resulting from the consumption of (herbal) teas was calculated as described before [20], as shown in Equation (1). The interim REP factors for each individual PA, which derived from the data of in vitro cytotoxicity and genotoxicity in *Drosophila* and acute toxicity in rodents (LD50) [12], were used to correct the PA concentrations.
(1)$$\mathrm{EDI}=\frac{\mathrm{Sum}\;\mathrm{of}\;\mathrm{concentration}\;\mathrm{of}\;\mathrm{each}\;\mathrm{PA}\;\mathrm{by}\;\mathrm{or}\;\mathrm{no}\;\mathrm{REP}\;\mathrm{correction}\;*\;\mathrm{daily}\;\mathrm{intake}\;\mathrm{of}\;\left(\mathrm{herbal}\right)\;\mathrm{tea}}{\mathrm{Body}\;\mathrm{weight}\;}$$
where the daily intake of (herbal) tea was estimated to be 2 g, which roughly corresponds to one cup of tea; REP factors were 1.0 for cyclic diesters and heliotridine-type (7S) open diesters, 0.3 for heliotridine-type (7S) monoesters, 0.1 for retronecine-type (7R) open diesters and 0.01 for retronecine-type (7R) monoesters (e.g., indicine, intermedine and lycopsamine) (Appendix A). A default adult body weight of 70 kg was used as suggested [23].

The MOE values for the chronic lifetime exposure to (herbal) teas were calculated as follows:(2)$$\mathrm{MOE}=\frac{{\mathrm{BMDL}}^{10}\mathrm{of}\;\mathrm{riddelliine}}{\mathrm{EDI}\;}$$
where the BMDL_10_ of riddelliine is 237 μg/kg bw/day; the MOE values for the short-term exposure were calculated based on Haber’s rule and a lifetime expectancy of 75 years, as described previously [21]. The MOE value being below 10,000 suggests a potential health risk related to the exposure that cannot be excluded and high priority might be given for risk management [3].

The maximum number of weeks that could result in an MOE of 10,000 based on the daily consumption of one cup of tea was calculated as Equation (3), according to the previous studies [3,14,15].
(3)$$\mathrm{The}\;\mathrm{maximum}\;\mathrm{number}\;\mathrm{of}\;\mathrm{weeks}=\frac{\mathrm{BMDL}10\;\mathrm{of}\;\mathrm{riddelliine}*75\;\mathrm{years}*52\;\mathrm{weeks}}{\mathrm{EDI}*\mathrm{MOE}}$$

All calculations above were based on an assumption that the concentrations reported are representative for the specific tea and that the exposure to PAs is exclusively due to that tea.

## 3. Results

### 3.1. PA Concentrations in (Herbal) Teas

In this study, a total of 68 individual PAs were analyzed, of which 23 PAs were not detected in all the investigated (herbal) teas (Appendix A). Seventeen PAs were found in lemon balm (melissa) tea, ranked first regardless of the REP correction, followed by tephroseris and lemon balm & liquorice (Table 2). None of the targeted PAs were present in citroen melisse and fresh peppermint tea. Most of the teas (92.5%) were detected with PAs, with the measured total levels varying from 13.4 μg/kg d.m. to 286,682.2 μg/kg d.m. (Table 2). In terms of the total PA content, tephroseris, borage and lungwort were the top three teas, which all originated from PA-producing plants. When taking the REP factors into account, the concentrations of total PAs were ranged from 1.3 μg/kg d.m. to 286,648.3 μg/kg d.m., which were generally lower compared to those measured levels in the (herbal) teas, except for green tea, gynura segetum and rooibos. It is of note that the total PA level for borage tea decreased by about 116.5-fold by the REP correction, amounting to 1440.6 μg/kg d.m. Whereas the PA concentrations in tephroseris were hardly altered by the REP correction, lungwort presented a considerable drop in its total PA level. This drop made the REP-corrected PA level for lungwort even lower than that for some teas from non-PA-generating plants, such as lemon balm, chamomile, rooibos and lemon verbena. Among the teas derived from non-PA-producing plants, lemon balm (melissa) and chamomile ranked first and second with their total PA levels, irrespective of REP correction.

The regulations of Germany and the Netherlands have indicated that the maximum limit for daily intake of 1,2-unsaturated PAs (including *N*-oxides) during a lifetime by a 70 kg person was 0.1 μg/day [24,25]. In this study, we calculated the daily intake of total PAs by consuming one cup of tea with and without REP correction. The results showed that there were nine types of teas resulting in the daily intake of PAs above 0.1 μg/day, regardless of REP correction (Table 2). These teas included tephroseris, borage, lemon balm (melissa), chamomile, eupatorium, rooibos, mix herb (1), lemon verbena and green tea. In addition, the daily intake of PAs due to the consumption of earl grey, lemon balm & liquorice, lungwort and sage & lemon myrtle could exceed the maximum limit when the REP factors were not applied.

According to the top three PAs and their concentrations, senkirkine and its congener neosenkirkine were the dominant PAs in the (herbal) teas from PA-containing plants except for borage and lungwort, while echinatine, retrorsine, integerrimine and senecionine as well as their *N*-ox congeners were frequently occurring in the non-PA-producing teas (Table 2). When taking the REP factors into account, senkirkine, neosenkirkine and petasitenine remained the same levels as their REP factors were derived to be 1 (Appendix A). In contrast, supinine, intermedine and their *N-*oxide congeners plus lycopsamine *N-*oxide correspond with a proposed REP value of 0.01, which could explain for the remarkable decreases in the total PA levels for borage and lungwort due to REP correction. Overall, the types of the top three PAs remained unchanged in most of the studied teas in response to REP correction, with one exception of lemon balm & liquorice, which was due to the REP factors of atropine and scopolamine, set at 0.

### 3.2. Estimated Daily Intake of PAs

Table 3 displays the EDIs for the studied (herbal) teas. With one cup of tea per day, the EDIs calculated based on the quantifiable levels of total PAs were ranged from 3.83 × 10^−4^ to 8.19 μg/kg bw/day. When using PA levels corrected by REP factors, EDIs were present in the range of 3.71 × 10^−5^ and 8.19 μg/kg bw/day. It should be noted that tephroseris tea had EDIs of 8.19 μg/kg bw/day, no matter whether REP correction was applied or not. Besides, borage tea exhibited an EDI of 4.80 μg/kg bw/day when REP factors were not considered. These EDI values exceed a “tolerable daily intake” (TDI) of 0.1 µg/kg bw/day for the total PAs in (herbal) preparations or extracts, which was provided by the Dutch national institute for public health and the environment (RIVM) [14]. The TDI was derived based on the NOAEL of 0.01 mg/kg bw/day for non-neoplastic changes due to the chronic exposure of riddelliine in rats and an uncertainty factor of 100, and therefore indicates an exposure level of PAs that may cause non-carcinogenic effects.

### 3.3. Risk Assessment for the (Herbal) Teas Based on Lifetime and Shorter Duration Exposure

The MOE values for the 21 types of (herbal) teas were evaluated according to two exposure scenarios, including the consumption of one cup of tea daily throughout the whole lifespan (Figure 1A) and shorter-than-lifetime during two weeks a year for 75 years (Figure 1B). Of the seven PA-producing plants-derived teas, tephroseris, borage and lungwort resulted in the low range of the MOE values between 29 and 4687 upon a lifelong daily consumption without REP correction. The MOE values for tephroseris and borage remained below 10,000 when taking REP factors into account, while that for lungwort was increased by 219,444 (Figure 1A). For tephroseris tea, even short-term consumption of two weeks/year resulted in a low MOE value of 752 regardless of REP correction (Figure 1B), which was well below 10,000, indicating that this tea may pose a potential risk for human health. Having borage tea two weeks per year resulted in an MOE value of 1285, contrasting with an MOE value well above 10,000 due to REP correction. Notably, lemon balm, a tea from non-PA-generating plants, showed MOE values just below 10,000 irrespective of REP correction when consumed daily for a lifetime. However, in the defined shorter duration exposure scenario, the resulting MOE values for this tea were well above 10000. Interestingly, since asteraceae, gynura segetum and heliotropium each contained a few PAs at low concentrations, use of these teas resulted in MOE values far above 10,000 regardless of the exposure duration and REP correction, although these teas were obtained from PA-containing plants. Similarly, consumption of citroen melissa, fresh peppermint, mix herb (2), mix herb (3), forest fruit tea and lemon balm & liquorice may not raise health concern as no PAs were found or the resulting MOE values for these teas were multiple orders of magnitude higher than 10,000. In addition, use of earl grey, chamomile, green tea, rooibos, sage & lemon myrtle and eupatorium derived MOE values over 10,000 irrespective of the exposure duration and REP correction, indicating no health concern.

### 3.4. Risk Assessment for the (Herbal) Teas Based on Shorter-Than-Lifetime Consumption

Providing that the number of weeks a year selected for a shorter-than-lifetime exposure has an influence on the MOE values and corresponding conclusion, the number of weeks during a 75-year lifetime that would cause an MOE of 10,000 was calculated in the current study. As shown in Figure 2, consumption of tephroseris tea for more than 11.2 weeks during a lifetime, which is corresponding to 0.1 weeks/year during 75 years, would already raise a concern, no matter whether the REP factors were applied or not. Having borage tea for up to 19.3 weeks during a lifetime (0.3 weeks/year) would be of little concern, whereas the number was increased by REP correction to 2245.6 weeks during a lifetime (29.9 weeks/year). Use of lungwort and lemon balm containing the PA concentrations as measured in this study would result in an acceptable exposure for 1828.0 and 3828.0 weeks during a life time, respectively (equivalent to 24.4 and 51.0 weeks/year, respectively). As the REP factors were applied, the maximal number of weeks for lemon balm tea was 3828.0 weeks, while use of lungwort would raise no concern for the whole lifespan. For the other types of teas, the number of weeks resulting in an MOE of 10,000 exceeded 3900 weeks, suggesting that use of these teas may not pose a health risk.

## 4. Discussion

Here, we investigated the PA levels in 21 types of (herbal) teas that were derived from both PA- and non-PA producing plants, based on which we performed a risk assessment using the MOE approach. In addition, we applied the REP factors for each PA that were analyzed to correct the PA concentrations and further calculated EDI and MOE for the (herbal) teas based on a lifetime and shorter duration exposure. Overall, the PA-containing plants-derived teas present a much wider range of the measured total PA levels from 13.4 to 286,682.2 μg/kg d.m., as compared to the levels ranging from 15.0 to 845.1 μg/kg d.m. for the teas from non-PA producing plants. This is in line with the findings reported by Griffine et al. (2014) [1] and Mulder et al. (2015) [13]. When taking REP factors into consideration, most of the samples showed decreased PA levels and, correspondingly, elevated the EDI and MOE values. Nevertheless, the daily consumption of tephroseris, borage and lemon balm (melissa) tea during a lifetime may pose a potential risk to human health.

One of the advantages in the present study is that a comprehensive set of 68 individual 1,2-unsaturated PAs was included, because we intend to reduce the risk of missing relevant PAs as much as possible. We found that PAs were occurring in 19 out of 21 types of the (herbal) teas, albeit that 23 PAs were absent from all the tested samples. Multiple studies on risk assessment for (herbal) teas were based on a set of 17–28 PAs [2,3,13]. Several PAs that were enriched in the (herbal) teas analyzed in this study had not been included in those previous studies, such as atropine, petasitenine, neosenkirkine, and integerrimine, supinine, echinatine and *N*-oxide isomers of these three PAs. It is conceivable that due to the omission of the major PAs, the reported total PA levels in the above-mentioned studies would be lower than those measured in the same type of (herbal) teas based on our method. For instance, the total PA level for borage tea has been shown to be 29,694 μg/kg d.m. based on the set of 28 PAs [3], while this amounted to be 167,846.6 μg/kg d.m. when using the set of abundant PAs in this study (Table 2). Of note, our data showed that supinine plus supinine *N*-oxide, which were usually omitted in previous studies, accounted for about 41% of the total PA levels for borage tea. Neosenkirkine was also missing in previous studies but it turned out to be the major PA contributor in five types of teas in the present study. A lower number of analyzed PAs and a lower analytical sensitivity have been implicated with a greater difference in the derived MOE values based on lower bound exposure estimates in the tea samples [3]. Therefore, it may suggest that the set of sufficient types of PAs should be included to improve the accuracy of the evaluation on the total PA levels occurring in (herbal) teas and the associated health risks.

It Is well accepted that the potency to induce toxicity may be different from one type of PA to another due to distinct metabolisms and the toxic effects of PA metabolites. To approach the improved accuracy of the risk assessment due to the exposure, we applied the REP factors for each individual PA that were analyzed and obtained the REP-corrected PA levels. By doing so, we found that the risk assessment for several teas was significantly affected by REP correction. For example, the MOE results indicated that a daily intake of lungwort and the consumption of borage for two weeks a year during a lifetime may pose a health risk, while, by REP correction, the same regime of tea consumption would be of no concern (Figure 1). It should be noticed that the interim REP factors used in this study were derived from the genotoxicity data and did not take some physiological conditions into account, e.g., the tumor formation and in vivo toxicokinetics [12]. This may compromise the accuracy of the risk assessment to some extent when using these REP values. Preferably, the REP factors used for the combined exposure to PAs should be derived from in vivo carcinogenicity potencies, which actually are largely lacking so far. In support of this notion, the Joint FAO/WHO Expert Committee on Food Additives (JECFA) and ESFA considered that the existing data are not sufficient to identify REPs for different PAs [20,26]. Hence, more researches aiming for actual carcinogenicity data that are capable of deriving REP factors for a large set of PAs should be fully encouraged in the future.

To facilitate the risk management for PAs in botanicals and botanical preparations, several organizations have established the regulation to define a maximum daily use and TDI for PA levels. For example, the BfR in Germany and RIVM in the Netherlands proposed a maximum limit for intake of PAs with 0.1 µg/day for a long-term (over six weeks) exposure scenario [24,25]. In the present study, the intake of PAs resulted from the daily consumption of nine types of (herbal) teas that exceeded the maximum limit set by the BfR and RIVM, even after REP correction (Table 2). It should be pointed out that an intake of PAs at 1 µg/day for a lifetime by a 70 kg person would result in an MOE of 16,600, which provides a sufficient safety margin. With respect to the non-cancer effects of PAs, a NOAEL of 0.01 mg/kg bw/day for hepatocyte cytomegaly was derived based on a long-term rat study [18,27]. According to that study and considering the safety factors, the RIVM provided a TDI of 0.1 µg/kg bw/day to indicate an exposure level of PAs that may cause non-carcinogenic effects. From a view of mechanism of action, the occurrence of pyrrole-protein adducts was thought to be the primary cause for PA-induced liver damage, as manifested in both humans and rodents in vivo studies [28,29,30,31,32,33]. Based on the REP-corrected EDIs calculated in our study, consuming one cup of tephroseris tea daily could be likely to induce non-cancer toxicity to human health. Altogether, our data suggest that the daily consumption of tephroseris tea may pose both carcinogenic and non-carcinogenic risks to human health, and thus a risk management may be needed.

It should be noted that multiple s”udie’ have been done with regards to risk assessment for PAs in (herbal) tea infusions. In 2017, the EFSA Panel on Contaminants in the Food Chain (CONTAM Panel) assessed the risks related to the presence of PAs in tea infusions as well as the other PA-containing foodstuff [20]. To ensure the actuality and robustness of EDI and MOE values for different age groups of the population, the CONTAM Panel employed at least six dietary surveys and over 60 observations per age group, as highly suggested before [34]. With the well-defined consumption survey data, the Panel was able to derive mean and the 95th percentile values of EDI and MOE for different types of teas consumed by the adult (referring to adults, the elderly and very elderly) and young (infants, toddlers and other children) population. For example, based on the chronic mean exposure levels, MOE values were ranged from 4300 to above 1,000,000 and from 1000 to over 1,000,000 for the adult and young population, respectively [20]. In this study, we mainly focused on the individual MOEs for (herbal) teas, assuming that one cup of tea per day is representative of most common behavior and there is no additional exposure to PAs from any other sources. One could argue that our approach may tend to underestimate the intake amounts of PAs and the related risks, since individuals may consume more (herbal) teas and/or add honey, a foodstuff that generally contains abundant PAs [34,35,36]. Indeed, apart from four teas that may raise a concern, the daily consumption of two cups of chamomile tea and three cups of rooibos tea during a lifetime would result in an MOE value lower than 10,000 based on the measured PA levels in this study (data are not shown). We did not include the young population in the risk assessment for a life-time exposure of PAs from (herbal) tea intake, as we argue that there is an overestimation when considering the fact that their body weights are remarkably increasing before they reach adulthood. It is also worthwhile to mention that we adopted the BMDL_10_ of riddelliine for the MOE calculation, which agrees with the proposal by the CONTAM Panel, resulting in additionally increased MOE values by a factor of 3.4 compared to that that for lasiocarpine of 70 μg/kg bw/day in previous studies [14,15,20,37,38]. Since the MOE approach was proposed by the EFSA in 2005, this approach has been either employed to perform a risk assessment for (herbal) teas or for comparison with the other methods [15,20,39,40,41,42].

It is challenging to perform a risk assessment for the combined exposure to PAs due to the intake of different types of (herbal) teas and other PA sources. For instance, a risk assessment report has shown that there are a group of subjects with high PA exposure due to the consumption of PA-containing teas and honey [20]. In addition, herb medicines, which generally contain a large amount of PAs, are used in a group of patients or during a specific period [21,43,44]. The evaluation of more complex scenarios, such as a shorter-than-lifetime exposure, is also a challenging issue in the field of risk assessment. It should be acknowledged that a dedicated survey on the consumption habits of (herbal) teas by the average population and by the 95th percentile population (heavy consumers) should be conducted and will contribute to risk assessment for the exposure to PAs.

## 5. Conclusions

In the present study, a comprehensive set of PAs in 21 types of (herbal) teas were analyzed using LC-MS/MS and the total PA levels were corrected by the REP factors for each PA. Based on these data, the risk assessment for (herbal) teas was performed using the MOE approach, assuming a daily consumption of one cup of tea. Most of the tea samples (92.5%) were detected with PAs. The measured total PA levels were ranged from 13.4 to 286,682.2 μg/kg d.m., while the levels were decreased by REP correction, ranging from 1.3 to 286,648.3 μg/kg d.m.. Senkirkine plus its isomer neosenkirkine were the dominant PAs in most of the (herbal) teas which were derived from PA-containing plants, while echinatine, retrorsine, integerrimine, senecionine and their *N*-ox isomers were the major PAs detected in the non-PA producing teas. In general, the PA levels in PA-producing plant-derived teas were higher than those from non-PA producing plants. Hence, efforts should be made to reduce or avoid the use of raw materials from PA-producing plants as tea components in terms of controlling PA intake.

The MOEs for PAs due to the intake of (herbal) teas for a life-time and during a short-term period of two weeks per year were calculated using the BMDL_10_ of riddelliine as the new PoD and 70 kg for the estimated adult body weight, amounting to values ranging from 29 to >1,000,000 and from 725 to >1,000,000, respectively. These values were increased by REP correction in most of the (herbal) teas. Despite this, our data indicate that daily consumption of tephroseris, borage and lemon balm (melissa) tea during a lifetime may raise a concern. In addition, shorter-than-lifetime exposure due to an intake of tephroseris tea would be a health concern and may pose a potential non-carcinogenic risk. Therefore, a priority for a risk management of tephroseris tea should be warranted.

To our best knowledge, this is the first study that applied the REP factors for risk assessment for PA exposure from (herbal) teas. In addition, 68 individual 1,2-unsaturated PAs were included to improve the accuracy of the assessment on associated health risks. The number of the analyzed PAs in our study is higher than the requirement of the ESFA as well as of that in the other studies [2,3,13,42,45,46]. On the other hand, there were some limitations in this study. For example, the lack of sufficient dietary survey data regarding tea consumption for different age population and in different scenarios leads to a compromised evaluation on the actual risks. Additionally, despite that applying Haber’s rule may provide a reasonable first approach for an MOE-based risk assessment for a shorter-than-lifetime exposure, this approach awaits to be validated [21]. Nevertheless, the results of the current study present the need for the development of a widely accepted method for assessing the risks of botanicals and botanical preparations containing genotoxic and carcinogenic compounds during a shorter-than-lifetime exposure.

## Data Availability

Data is contained within the article or Appendix A.

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
