# Peer review of "Risk Assessment of (Herbal) Teas Containing Pyrrolizidine Alkaloids (PAs) Based on Margin of Exposure Approach and Relative Potency (REP) Factors"

_foods, 2022, doi:10.3390/foods11192946_

Round 1

Reviewer 1 Report

The submitted article ‘Risk assessment of (herbal) teas containing pyrrolizidine alkaloids (PAs) based on margin of exposure approach and relative potency (REP) factors’ is very interesting and professional article. This is a kind of rare studies, hence is very desire.  In my opinion manuscript is written correctly, concisely in most parts (some mistakes and remarks should be corrected). Below I pointed most of mistakes and matters for explanation.

1.      In introduction the emphasis should be placed on justification

2.      In introduction you should include chemical structure of analysed PAs

3.      The good idea should be a summary as a graphical workflow in the introduction.

4.      Perhaps you can add more details for analyzed teas?  Including Code of sample, form,  amount of raw material for infusion process, eg; time for infusion process (brew time), country of origin, EAN.

5.      Estimated daily intake (EDI) and MOE calculation should be based on actual EFSA requirements – please compare with https://doi.org/10.2903/j.efsa.2017.4908 and emphasize this in manuscript (you cited, but it is important document that you should mentioned in the text, not only as citation because it is relevant from regulatory point of view)

6.      Conclusion part should be more informative (more details, perhaps in brackets?)

7.      Please include advances and disadvantages of your studies in conclusions

I totally agree that this is very important subject, and it is very important for publication in MDPI. I will recommend this article for publication in FOODS after minor revision.

Reviewer 2 Report

The manuscript presents the risk assessment of herbal teas containing pyrrolizidine alkaloids (PAs), using a new method based on REP factors .

The following are some comments and suggestions for authors:

1. Lines 113-114 It is important to explain why the elution of PAs from black and green teas is different

2. Materials and methods In total were analyzed 21 types of commercial teas/147 samples but the repeatability range is missing

3. The manuscript lacks in-depth discussion, as there are many papers on herbal teas PAs risk assessment. Also, this will increase the number of references, which is very low.
